evolution/genetics

Neanderthal, introgression, human, D statistics, heterozygosity, mutation rate

**Author for correspondence:**
William Amos
e-mail: w.amos@zoo.cam.ac.uk

# Signals interpreted as archaic introgression appear to be driven primarily by faster evolution in Africa

## William Amos

Department of Zoology, Downing Street, Cambridge CB2 3EJ, UK

 WA, 0000-0002-0971-9914

Non-African humans appear to carry a few per cent archaic DNA due to ancient inter-breeding. This modest legacy and its likely recent timing imply that most introgressed fragments will be rare and hence will occur mainly in the heterozygous state. I tested this prediction by calculating D statistics, a measure of legacy size, for pairs of humans where one of the pair was conditioned always to be either homozygous or heterozygous. Using coalescent simulations, I confirmed that conditioning the non-African to be heterozygous increased D, while conditioning the non-African to be homozygous reduced D to zero. Repeating with real data reveals the exact opposite pattern. In African–non-African comparisons, D is near-zero if the African individual is held homozygous. Conditioning one of two Africans to be either homozygous or heterozygous invariably generates large values of D, even when both individuals are drawn from the same population. Invariably, the African with more heterozygous sites (conditioned heterozygous > unconditioned > conditioned homozygous) appears less related to the archaic. By contrast, the same analysis applied to pairs of non-Africans always yields near-zero D, showing that conditioning does not create large D without an underlying signal to expose. Large D values in humans are therefore driven almost entirely by heterozygous sites in Africans acting to increase divergence from related taxa such as Neanderthals. In comparison with heterozygous Africans, individuals that lack African heterozygous sites, whether non-African or conditioned homozygous African, always appear more similar to archaic outgroups, a signal previously interpreted as evidence for introgression. I hope these analyses will encourage others to consider increased divergence as well as increased similarity to archaics as mechanisms capable of driving asymmetrical base-sharing.

# 1. Introduction

The current dogma is that humans inter-bred widely with other archaic hominins such as Neanderthals and Denisovans [1–7] apparently more or less whenever the species' ranges overlapped [7–9]. The discovery of hybrid skeletons [9,10] appears to lend unquestionable support to the fact that inter-breeding did occur. There is also evidence that archaic DNA has been selected and thereby facilitated adaptation [11,12], with Denisovan haplotypes found in Tibetans living at high altitude [13] and evidence for selection against Neanderthal alleles of meiotic genes [14]. Over time, the length of introgressed fragments appears to get shorter [15], consistent with the breakdown of fragments over time through recombination.

Several studies have inferred the distribution of archaic DNA in modern humans, generating introgression landscapes [6,16] as well as estimates for the overall amount of archaic DNA in different global populations [3,6]. Across the genome, regions lacking any evidence of introgression appear to be rare [16] though may make up as much as a third of the genome. Introgression peaks appear broadly similar between Europe and East Asia [16]. In terms of quantity, Neanderthal legacies are rather uniformly distributed across Eurasia [17], but increase significantly west to east [18]. By contrast, Denisovan legacies reach 5% or more in Papua New Guinea and Australia [3,19], but are substantially lower elsewhere.

Although some introgressed fragments may have been selected to high frequency, the majority are expected to be rare. Consequently, most Neanderthal alleles will occur in heterozygote genotypes. If so, filtering sites according to whether they are heterozygous or homozygous should have a major impact on statistics used to quantify asymmetric base-sharing between an outgroup such as Neanderthals and one of two humans/human populations. The most widely used statistic is called D and focuses on sites with two alleles, labelled A and B, in four taxon alignments: human1, human2, archaic = B and chimpanzee = A [4,20,21]. When the human alleles differ, two states are possible, ABBA and BABA, and D is the normalized difference in counts, calculated as (nABBA − nBABA)/(nABBA + nBABA), where n signifies 'number of'. Significant asymmetry is classically interpreted as evidence of introgression. Note that the sign of D reverses when the positions of the two humans are switched but that in the classic comparison between an African and a non-African, D is always calculated such that a positive value indicates excess 'B' alleles in the non-African. Under the introgression model, D should reduce to near-zero if sites where non-Africans are heterozygous are excluded. To test this prediction, I calculated D between all pairs of individuals in the 1000 genomes Phase 3 data, conditioning variously on whether all sites in a second individual are heterozygous, $D_{ANY,HET}$, unconditioned, $D_{ANY,ANY}$, or only allowed to be homozygous, $D_{ANY,HOM}$ (the first individual can also be conditioned, but I focused on rotating population identity). I find that in simulated data, conditioned D performs exactly as expected, with exclusion of heterozygous sites in non-Africans killing the signal. However, applied to real data, conditioning reveals the exact opposite pattern, with the signal being driven by heterozygous sites in Africans acting to increase divergence from the archaic.

# 2. Results

## 2.1. Simulated data

I began by using coalescent simulations to confirm the intuitive expectation that signals of introgression will be associated largely with sites that are heterozygous outside Africa. With the coalescent simulator MS [22], I simulated a simple scenario in which hominins splits from the chimpanzees 6 million years ago, Neanderthals split from humans 300 000 years ago and the human lineage splits into Africans and non-Africans 70 000 years ago. Non-Africans suffer a strong bottleneck immediately after they leave Africa and receive a 3.4% pulse of Neanderthal chromosomes 50 000 years ago. Bottleneck intensity and introgression were adjusted to create realistic values for heterozygosity lost (25%) and D (approx. 4%). Each of 80 000 independent simulation runs generates the equivalent of 25 Kb of non-recombining sequence. From these data, I calculated six versions of D, always calculated such that positive values indicate introgression of Neanderthals into simulated non-Africans:

1. using all sites;
2. using only sites that are homozygous in the simulated African;
3. using only sites that are heterozygous in the simulated African;
4. using only sites that are homozygous in the simulated non-African;

5. using only sites that are heterozygous in the simulated non-African; and

6. using only sites that are homozygous in both individuals.

These six versions yield D values of 3.6%, 5.3%, 1.3%, 1.5%, 8.2% and 1.3%, respectively, consistent with expectations. The largest D is obtained using only sites that are heterozygous in non-Africans and the minimum values are obtained from sites that are either homozygous outside Africa or heterozygous inside Africa. Conditioning on Africans being homozygous increases D because it enriches for sites that were fixed for the ancestral allele in humans prior to non-Africans receiving derived Neanderthal alleles. Similarly, conditioning on sites being heterozygous in Africa reduces D because in these infinite allele simulations all such sites are generated by incomplete lineage sorting. Varying population sizes and split timings increase or decrease these values but the rank order of D values is robust. Repeating this analysis without introgression yields D values that are all indistinguishable from zero (no value greater in magnitude than 0.005), showing that conditioning does not create non-zero D on its own, it merely acts to modulate patterns already in the data.

I also explored how conditioned D performed *within* each simulated human population. Within Africa, all conditioned D values are approximately zero, ranging from −0.1% to 0.1%, confirming that conditioning alone does not generate large D values unless there is an asymmetry to be exposed. By contrast, within simulated non-Africans, D is large, magnitude 5.7% to 5.8%, with positive and negative values always indicating that individuals carrying the greater number of heterozygous sites are closer to the simulated Neanderthal outgroup. This is exactly what is expected, because heterozygous sites are enriched for rare alleles, some of which come from introgression.

## 2.2. Analysis of the 1000 genomes data

Having verified that, under a model based on introgression, non-zero D is driven mainly by sites that are heterozygous outside Africa, I next repeated the six D calculations using the 1000 genomes data [23]. All comparisons were made between all pairs of individuals and then averaged over each of all possible inter- and intra-population combinations. To make the analysis more general and not focused purely on African–non-African comparisons, I calculated D(*P1*, *P2*, *Neanderthal*, *Chimpanzee*), where *P1* and *P2* are two individual humans drawn from the same or from different populations, and the six measures were taken as:

1. using all sites;

2. using only sites that are homozygous in the *P1* individual;

3. using only sites that are heterozygous in the *P1* individual;

4. using only sites that are homozygous in the *P2* individual;

5. using only sites that are heterozygous in the *P2* individual; and

6. using only sites that are homozygous in both individuals.

In every comparison where the two individuals were from different populations, the individual from the population with the lowest index number was placed in position *P1* (indexes are: 1–5 = Europe; 6–10 = East Asia; 11–15 = South Asia; 16–22 = Africa; 23–26 = America). This ensures that all inter-population comparisons are made in the same direction. As an additional check, I also coded the calculation of D based on a probabilistic approach, using genotype frequencies in each population to calculate the expected frequencies of each possible two-genotype combination (electronic supplementary material, table S1). Essentially identical results were obtained. Figure 1 summarizes the results for all six measures, with strong colours indicating large magnitude D values and blue and pink indicating positive and negative D, respectively. With no conditioning, the classic pattern is recovered, with all comparisons between Africans and between non-Africans being near-zero, and all African–non-African comparisons yielding large D values with the sign of D indicating that non-Africans are closer to Neanderthals. A similar but much-diluted pattern is seen when the data are conditioned such that all sites are homozygous in both individuals, emphasizing the way the signal is carried mainly but not exclusively by heterozygous sites.

When one and only one individual is conditioned while the other is unconditioned, a similar but importantly different pattern is found. As before, large D values invariably involve at least one African individual. However, wherever there is one and only one African individual, as long as this individual is conditioned always to be homozygous, D is always near-zero. In addition, D is now large *within* Africa, both within individual African populations and in comparisons between

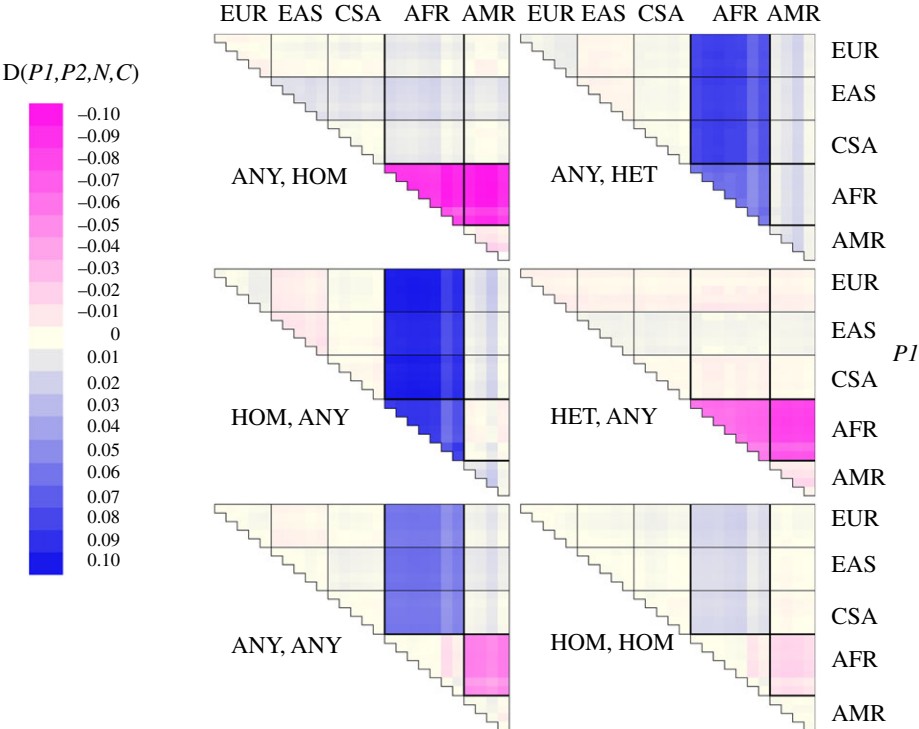

**Figure 1.** Variation in D between human populations and its dependence on whether sites are heterozygous or homozygous. All D values were calculated between pairs of individuals as D(P1,P2,N,C), where P1 and P2 are the two humans, arranged such that the population index (see Methods) of P1 is always less than or equal to the index of P2. The six panels represent different conditioning: thus, ANY, HOM indicates that only sites where P2 is homozygous are included, P1 being unconstrained. D values are averaged across all possible combinations within and between all populations / population combinations and colour coded by their mean value, varying from −0.1 (dark pink) through to 0.1 (dark blue). Actual values are tabulated in electronic supplementary material, table 1. The population/region of P1 is defined by the row and of P2 by the column. Thus, in the top left panel 'ANY, HOM', the dark pink block indicates that D is large and negative whenever P1 is African and P2 (constrained to be homozygous) is either African or American, all other D values being close to zero.

individual Africans from different populations. This is the exact opposite of what is seen in the simulated data, where conditioned D is large outside Africa and near-zero inside. Since it is inconceivable that individuals within the 1000 genomes are ordered according to the size of their archaic legacy, the strong implication is that, within Africa, individual identity is largely irrelevant and large D-values are driven more or less entirely by the conditioning process. To confirm this speculation, I chose one African population (ESN) and, for each pair of individuals, calculated both $D_{ANY,HOM}$(African1, African2,N,C) and $D_{ANY,HOM}$(African2,African1,N,C). The resulting values are dominated by the conditioning process, all values being strongly negative regardless of the order of the individuals. Despite this, the paired values do exhibit a weak but highly significant negative correlation, consistent with a much weaker, order-dependent signal also being present (figure 2).

When both individuals are heterozygous, by force the same numbers of ABBAs and BABAs are always generated (remember, all comparisons are between individuals, not population samples). For this reason, D is always zero if both individuals are conditioned to be heterozygous and this combination was therefore not included. Moreover, when one individual is conditioned to be heterozygous and the other individual is unconditioned, the outcome is very similar in effect to conditioning the second individual to be homozygous. This is because, as explained, sites where both individuals are heterozygous effectively cancel each other out, acting merely to increase ABBA and BABA counts equally.

## 2.3. Are back-mutations sufficiently common?

Previous studies generally assumed that the only possible source of asymmetrical ABBA and BABA counts is introgression and hence infer introgression wherever the null hypothesis of symmetric ABBA

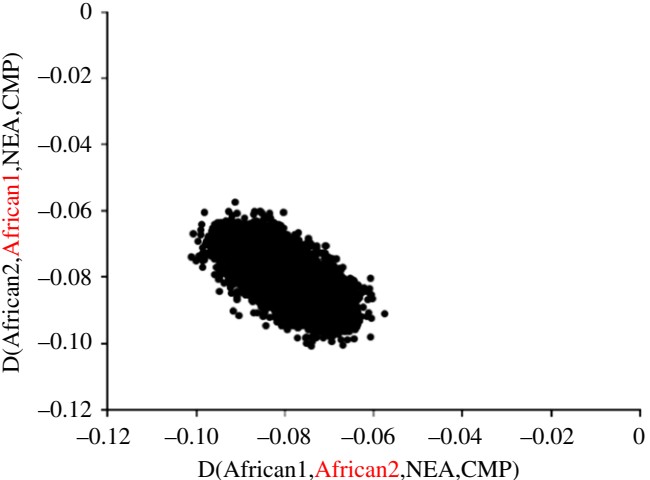

**Figure 2.** Conditioned D depends very little on individual identity. Within Africa, D always has large magnitude when one individual is conditioned. To confirm the implication that individual identity has minimal impact relative to the conditioning process, I considered all pairwise comparisons between individuals within one randomly selected African population, ESN, and for each comparison calculated $D_{ANY,HOM}$(African1,African2,N,C) and $D_{ANY,HOM}$(African2,African1,N,C), where N and C are the Neanderthal and chimpanzee, respectively. Note that all D-values are strongly negative even though a residual signal not due to conditioning is sufficient to drive a negative correlation, indicating that the conditioned homozygous individual is closer to the Neanderthal. In the axis legends, the conditioned homozygous individual is emphasized in red.

and BABA counts can be rejected. I now add a second alternative hypothesis, namely, that back-mutations convert the common BBBA state to ABBAs and BABAs in a system where mutation rate differs between human populations. Since introgression cannot account for the way conditioned D unambiguously identifies heterozygous sites in Africans as the primary driver of large D in humans, the implication is that back-mutations are far commoner than has been previously assumed. To test this prediction, I attempted to estimate the likely frequency of back-mutations by counting SNPs with three alleles [24]. Here, at least two mutations must have occurred, including one transversion. Back-mutations are 'silent' because they create the original ancestral allele, but can reasonably be assumed to occur about twice as often as triallelic SNPs are generated (two transitions are approximately twice as likely as one transition and one transversion [25]). Triallelic SNPs are coded 'MULTI-ALLELIC' rather than 'SNP' in the 1000 genome data and are often ignored, but I counted 257 827 occurrences across all autosomes, implying over half a million sites carrying back-mutations. Moreover, this is probably an underestimate because the 1000 genomes data are low coverage and rely on extensive imputation which will often cause rare third alleles to go undetected. Equally, conservative curation will tend to remove third alleles that lack strong support. Note, triallelic sites are unlikely to be generated mainly by sequencing errors because only 1% of these sites carry a singleton as the rarest allele. This analysis is not intended to provide an accurate estimate of the back-mutation rate, but instead simply to demonstrate that large numbers of back-mutations do exist to the extent that models of evolution that rely on back-mutations occurring in appreciable numbers should not be dismissed *a priori*.

## 3. Discussion

I introduce a new measure, conditioned D, that aims to uncover the origin of excess base-sharing with archaics by resolving where the predominant signal lies in terms of heterozygous and homozygous sites in each population. I began by testing the method on simulated data. When there is no introgression, conditioning never produces non-zero D, showing that there is nothing inherent in the conditioning process that generates artefactual signals. Applied to data simulating Neanderthal introgression outside Africa, conditioning behaves as expected, greatly reducing D when heterozygous sites outside Africa are excluded and increasing D when only heterozygous sites outside Africa are included. In comparisons between simulated non-Africans, conditioning one individual either to be always heterozygous or always homozygous produces large D, unmasking the signal of introgression that is not seen otherwise. The sign of D always agrees with the expectation that the individual with

more heterozygous sites appears closer to the archaic. By contrast, comparisons between pairs of simulated Africans never reveal large D, regardless of conditioning. These simulations confirm that conditioning acts as expected, by modulating signals of recent introgression by changing the proportion of heterozygous sites that each individual contributes to D. In this way, conditioning is able to expose underlying signals even within a single population but, importantly, conditioning does not generate spurious signals unless an underlying signal is already present.

Applying conditioning to the 1000 genomes data reveals a picture that is almost the exact opposite of the simulated data. The within-region comparisons are particularly informative, with all conditioned comparisons within Africa yielding large D and all conditioned comparisons between pairs of non-Africans yielding near-zero D. By implication, non-Africans do not carry an appreciable signal that can be unmasked and instead almost the entire signal resides inside Africa. This picture is strengthened by African–non-African comparisons, where conditioning the African to be homozygous results in near-zero D, abolishing the classical result. Heterozygous sites in Africa thus play a pivotal role in driving D, an observation that suggests a simple model capable of explaining the entire picture. Individuals can be placed in three main classes according to proportion of sites they carry that are heterozygous in Africa, in decreasing order: $African_{HET}$ > African > [$African_{HOM}$ and non-African]. Wherever individuals are compared from the same class, D ~ 0, and wherever individuals belong to different classes, the individual from the lower class (fewer heterozygous African sites) appears closer to the archaic. This model explains all the panels with one apparent exception, $D_{ANY,HET}$(African,non-African,N,C) is near-zero. Here, heterozygous sites in the unconditioned African might be expected to drive large D. However, by conditioning on the non-African sites always being heterozygous, the signal is exactly cancelled: in sites where both individuals are heterozygous, equal numbers of ABBAs and BABAs are generated (the frequency of the putative Neanderthal allele is 0.5 in both individuals). Put another way, an African signal is only generated by sites where the African is heterozygous and the non-African is homozygous ancestral, and all such sites are excluded by the conditioning process.

Recent studies suggest that Africans may carry more Neanderthal DNA than previously suspected [26], a possibility that might help to explain why conditioned D is large inside Africa. However, for a number of reasons this seems unlikely. First, the sign of D is 'wrong'. Under an introgression model, heterozygous sites cause an individual to appear closer to the archaic but in every instance I find that heterozygous sites in Africans cause the individual to appear more distant. Second, even if the presence of introgressed fragments in Africa did contribute to non-zero D, this does not explain why conditioned D remains resolutely close to zero outside Africa where many introgressed fragments have been inferred [3,4,17]. This absence is particularly puzzling given that a strong signal is present in simulated data that include introgression outside Africa.

The model suggested by the data identifies alleles carried in the heterozygous state by Africans as the primary drivers of non-zero D. Specifically, the sign of D indicates that the individual carrying relatively more African heterozygous sites is invariably less related to the archaic than the individual they are being compared against. Since the probability of an allele being found in the homozygous state increases with allele frequency, heterozygous sites will be strongly enriched for rare and hence recently evolved variants [27]. Putting these elements together suggests a model where large D is driven by a higher mutation rate in Africans causing relatively greater divergence from Neanderthals. Individuals not carrying heterozygous African sites, or who carry fewer than the individual against whom they are being compared, therefore appear closer to the ancestral state and, hence, closer to related taxa such as Neanderthals.

A model where Africans are unusually different from Neanderthals through accelerated divergence rather than non-Africans being unusually similar to Neanderthals though carrying introgressed fragments requires both a large number of back-mutations and variation mutation rate between human populations. Specifically, the mutation rate in Africa would have to have been higher than the mutation rate outside Africa *since the out of Africa event*, causing significantly more back-mutations in Africans. By counting triallelic sites, I show that very large numbers of back-mutations are indeed present, estimated at more than half a million. A small excess mutation rate outside Africa has also been reported [17] but this goes in the 'wrong' direction and was based on all variants, so the analysis will be dominated by commoner variants that arose before the out of African event. A previous study using very similar methods but which only looked at rare variants that mostly arose *after* the out of Africa event, found the exact opposite pattern, with mutation rate being appreciably higher in Africa [28]. This result is consistent with the proposed model but more work is needed to ascertain whether the effect size is sufficiently large.

Further evidence that mutation rate may vary between human populations comes from studies of mutation type, with Africans and non-Africans showing highly significant differences in the relative rates at which different base triplets mutate [29–31]. This variation in rate has since been linked to variation in flanking sequence heterozygosity, with some types of mutation increasing in probability as flanking heterozygosity increases, while others decrease [32]. Consequently, the large loss of heterozygosity that occurred out of Africa [33] would have caused some mutations to increase and others to decrease in rate, with any (likely) imbalance manifest as a change in overall rate.

Current results appear to contradict a large and increasing body of evidence supporting the presence of archaic sequences in modern non-African genomes. Reconciling the two pictures will take further work. However, it should be noted that most methods used to infer the presence of introgressed sequences make the explicit assumption that mutation rate is constant [4,16,20]. Consequently, any rejection of the null hypothesis that two human sequences are always equidistant from an archaic outgroup is interpreted as evidence of introgression, with the sequence closer to the archaic being inferred to carry introgressed DNA. My analysis reveals strong support for an alternative model whereby, instead of one human sequence being closer to an archaic because it carries introgressed archaic segments, the second individual appears more diverged from the archaic due to a higher mutation rate. Distinguishing between these two mechanisms is not easy and, to my knowledge, appears not to be included in previous studies. Some of the strongest evidence for a legacy comes from the identification of introgressed haplotypes, but, putting aside the extended regions found in genuinely hybrid skeletons [9,10], interpreting these results also becomes challenging if there is appreciable variation in mutation rate between populations. This problem is compounded by recent reports that human mutations often occur simultaneously in clusters of up to at least eight substitutions, all in a single event and almost invariably all on the same chromatid [34]. Such clusters will be identified as haplotypes by any simple algorithm and will tend to be commoner in populations with a higher mutation rate.

Finally, the question arises as to whether there are other possible explanations. Ultimately, significant asymmetric base-sharing with an archaic can be created by only three mechanisms: introgression, population sub-structure and back-mutations coupled with variation in mutation rate between populations. Population sub-structure operates through variation in relatedness between Neanderthals and different African populations prior to the out of Africa event [35–37]. The current analysis appears to negate simple models based on hybridization: if most hybridization occurred outside Africa, heterozygous sites in non-Africans should carry most of the signal but they do not, while if hybridization occurred mainly inside Africa, this would explain the dominant role of heterozygous sites in Africans but then fails because overall D would be negative. Ancient population sub-structure inside Africa may play a role and this is not addressed by conditioned D, though it should be remembered that the older the process the more time key alleles will have had to drift and hence the weaker will be the link between heterozygous sites and D.

In conclusion, I present a simple analysis that reveals an unexpected pattern in which non-zero human D statistics are unambiguously dominated by heterozygous African genotypes. These sites invariably cause the African to be less closely related to archaics and so appear to carry signatures of increased divergence from our common ancestor. More work is needed to reconcile these results with those of previous studies that conclude most non-African humans carry 1–2% archaic sequences. The conditioning method appears to provide a useful new tool, capable of exposing and enhancing signals even within a single population.

# 4. Methods

## 4.1. Data

Data were downloaded from Phase 3 of the 1000 genomes project [23] as composite vcf files, one for each chromosome (ftp://ftp.1000genomes.ebi.ac.uk/vol1/ftp/release/20130502/). These comprise low-coverage genome sequences for 2504 individuals drawn from 26 modern human populations spread across five geographic regions: Europe (GBR, FIN, CEU, IBS, TSI); East Asia (CHB, CHS, CDX, KHV, JPT); Central Southern Asia (GIH, STU, ITU, PJL, BEB); Africa (LWK, ESN, MSL, YRI, GWD, ASW, ACB) and the Americas (MXL, CLM, PUR, PEL). Populations are listed in the order they appear in the analysis (GBR = 1, FIN = 2 ⋯ PEL = 26). Human–chimpanzee alignments at http://www.ensembl.org/info/data/ftp/ were downloaded and aligned to hs37d5. To minimize alignment issues, bases

within 30 either end of a contig. were excluded. The resulting files are available on Dryad. Individual chromosome vcf files for the Altai Neanderthal genome were downloaded from http://cdna.eva.mpg.de/neandertal/altai/AltaiNeandertal/VCF/. For analyses presented here, I focused only on homozygote archaic bases, accepting only those with 10 or more reads, fewer than 250 reads and where greater than 80% were of one particular base. This approach sacrifices modest numbers of (usually uninformative) heterozygous sites but benefits from avoiding ambiguities caused by coercing low counts into genotypes.

## 4.2. Data analysis and simulations

All analyses of the 1000 genomes data and archaic genomes were conducted using custom scripts written in C++ using Visual Studio 2013. The central analysis that calculates conditioned D was conducted twice, once based on all pairwise comparisons between individuals and once using a probabilistic approach in which the frequency of each genotype combination was inferred from their population frequencies. Both methods yield essentially identical results and the simpler, faster probabilistic code is annotated and available in electronic supplementary material, S1. The same code with very minor adjustment can be used to count triplet sites. Simulated data were generated using the coalescent program ms [22]. The base model was coded:

```
./ms 202 100000 -t 10 -I 4 1 1 100 100 -ej 6 2 1 -ej 0.3 3 2 -es 0.05 4 0.96 -ej 0.0501 5 2 -ej 0.07 4 3 -en 0.0685 4 0.007
```

I assume a haploid population size of 10 000, a mutation rate of $10^{-8}$ and set theta to 10, such that each of 100 000 non-recombining fragments is 25 Kb long. The hominin–chimpanzee split is taken to be 6 000 000 years ago, the Neanderthal–human split 300 000 years ago (generation length = 25 years) and the out of Africa event 70 000 years ago. With these parameter values, an introgression event of 3.4% at 50 000 years ago generates a realistic D of approximately 5%. A bottleneck occurs immediately post 'out of Africa' with parameters set to cause a realistic average loss of 25% of initial heterozygosity.

Data accessibility. The analysis is based on publicly available data from the 1000 genomes project. The example code used to conduct these analyses along with the raw output values on which the main figure is based has been uploaded as part of the electronic supplementary material. Data are available at the Dryad Digital Repository: https://datadryad.org/stash/share/ichHKrWj7hqlznOaR6NQVzITgp40dlqWvWA gAxyafiQ [38].

Competing interests. This publication is a continuation of work that formed the basis of a public challenge presented in the form of a cash bet. The public challenge was a five-figure cash bet offered from 2017 following the posting of a related preprint (https://www.biorxiv.org/content/early/2017/05/03/133306) to encourage the community to engage with the research questions posed. At the time of acceptance of the paper, details of the challenge were available at my website https://www.zoo.cam.ac.uk/directory/william-amos and https://www.researchgate.net/project/Neanderthal-introgression-a-case-of-smoke-and-mirrors. However, the current paper has no overlap with this earlier preprint and the challenge itself expired in June 2019.

Funding. This work was not funded.

Acknowledgements. I am indebted to Simon Martin for many useful discussions and particularly for saying 'I think I'd believe you if you get the same result after constraining all sites in P1 to be homozygous'! Constructive comments from a Referee helped improve clarity and accessibility.

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
