## [Reviewer comments · Royal Society Open Science]

Review History

RSOS-191900.R0 (Original submission)

Review form: Reviewer 1

Is the manuscript scientifically sound in its present form?

No

Are the interpretations and conclusions justified by the results?

No

Is the language acceptable?

Yes

Do you have any ethical concerns with this paper?

No

Have you any concerns about statistical analyses in this paper?

Yes

Recommendation?

Major revision is needed (please make suggestions in comments)

Comments to the Author(s)

The author calculates D statistics between African and non-African modern humans, neanderthal and chimp, both on data simulated to match inferred population histories and on real data from 1001/neanderthal genome projects. When doing this, the author uses different combinations of conditioning of modern human samples on being homozygous or heterozygous at the considered variants. The author observes a strong qualitative mismatch between the results from the simulations that include neanderthal->non-African introgression and the real data and uses this to argue that this introgression event cannot explain the data. He suggests that the data is better explained by elevated mutation rates in African human populations.

First, I want to emphasise that the D-statistic as defined by Durand et al. 2011 MBE and Patterson et al. 2012 Genetics tests for significant asymmetry in four taxon topologies, as is expected under gene flow between non-sister lineages. However, the actual magnitude of the D-statistic is not easily comparable between different demographic scenarios or across tests. Hence, drawing conclusions from comparing the magnitude of different D-statistics is problematic. Connected to that, conditioning on different types of segregation patterns in the samples/populations used in the test changes the expectation of D from 0 in the absence of introgression (or other processes that could lead to $D \neq 0$) to some other value. The author is of course aware of this and bases his points on unexpected deviations from 0 when doing different conditioning. However, it is not something the D-statistic is designed for and conclusions based on that have to be interpreted with care.

That being said, the findings by the author are unexpected, and if true interesting and worth being explored. However, what makes me skeptical about the findings is all the results shown in Fig. 1 would make perfect intuitive sense if the conditioning "ANY, HOM" etc. would mean $P1=HOM$, $P2=ANY$, rather than the other way round as it is interpreted in the text and described in the figure caption.

Such an explanation describes the data much better than the interpretation proposed by the author: That the signals are driven by increased mutation rate in Africans. In particular, only back mutations to the ancestral (chimp) state of a SNP in Africans could drive such a signal. I do think that different evolutionary rates can indeed bias Dstatistics, but, firstly, I am not sure about the intuitive explanation how this depends on conditioning, and, secondly, I do not believe that this would lead to such a strong effect among these rather recently diverged populations.

To me the most likely explanation is that the conditioning was just swapped by mistake. If this were the case then each single plot is as expected. Now they all show exactly the pattern opposite of what is expected (except for ANY/ANY and HOM/HOM were a swap would not matter). Even if we assume that there was no introgression, the at considered divergence scales segregating shared polymorphism will have much more importance than new back mutations for any realistic mutation rate. Hence, even assuming there was no introgression, the author would need to give an explanation how the expected effect of conditioning on the contribution of shared polymorphism to D-statistic, which would go into the precise opposite direction of what is observed could be totally overshadowed by back mutations.

To further elaborate on this, the expectation of D-statistics under conditioning is the following: The D-statistics is also called ABBA-BABA test, as it counts sites where you see two different alleles in P1 and P2 and two different alleles in (in this case) neanderthal and chimp. Per definition the chimp allele is called A and the neanderthal allele B.

If one conditions on P1 or P2 being heterozygous A/B then the sample being conditioned, per construction, carries the neanderthal allele and is thus expected to be closer to neanderthal. This is true even in the absence of any gene flow, if all ABBA/BABA patterns were due to ancestrally segregating polymorphism.

Hence, even in the absence of introgression, we would always expect the sample being conditioned to be het to be closer to neanderthal, or, conversely, the sample not being conditioned to be homozygous.

The author is exploring this to some degree with simulations, but I am missing simulations without gene flow and all the same comparisons as in fig. 1. For example, also considering cases with two African samples in the simulations. This would be very easy to do. My prediction is that the author would get qualitatively the same results as shown in Fig. 1 for ANY/HET and ANY/HOM if the population being conditions is swapped, i.e., for HET/ANY and HOM/ANY, etc. Only for ANY, ANY and HOM,HOM should D-statistics should be non-significant.

There needs to be a convincing explanation of how back mutations could lead to exactly to opposite observation of what is expected in a model without gene flow. What is the expected frequency of back mutations compared to segregating polymorphism?

At this stage I think it needs to be clarified whether the results are not simply due to a swap between the conditioned populations for ANY,HOM etc., which, as far as I can see, would perfectly explain all observations.

I will have further comments on more specific points but I do think the point above needs to be verified before going any further.

Decision letter (RSOS-191900.R0)

17-Jan-2020

Dear Dr Amos,

The editors assigned to your paper ("Signals interpreted as archaic introgression are driven primarily by accelerated evolution in Africa.") have now received comments from reviewers. While the reviewer states that the work is potentially of considerable interest, the reviewer raises a very substantive concern over the analysis that needs to be addressed in detail. We would like you to revise your paper in accordance with the referee and Associate Editor suggestions which can be found below (not including confidential reports to the Editor). Please note this decision does not guarantee eventual acceptance.

Please submit a copy of your revised paper before 09-Feb-2020. Please note that the revision deadline will expire at 00.00am on this date. If we do not hear from you within this time then it will be assumed that the paper has been withdrawn. In exceptional circumstances, extensions may be possible if agreed with the Editorial Office in advance. We do not allow multiple rounds of revision so we urge you to make every effort to fully address all of the comments at this stage. If deemed necessary by the Editors, your manuscript will be sent back to one or more of the original reviewers for assessment. If the original reviewers are not available, we may invite new reviewers.

- Data accessibility

If you wish to submit your supporting data or code to Dryad (<http://datadryad.org/>), or modify your current submission to dryad, please use the following link:
<http://datadryad.org/submit?journalID=RSOS&manu=RSOS-191900>

- Competing interests

- Authors' contributions

- Acknowledgements

- Funding statement

on behalf of Professor Peter Visscher (Associate Editor) and Steve Brown (Subject Editor)
 openscience@royalsociety.org

Associate Editor's comments (Professor Peter Visscher):

Associate Editor: 1

Comments to the Author:

The Editors have at this stage received comments from a single referee. Given the nature of those comments we believe it is most efficient if the author is given the opportunity to revise the paper now. In particular, the referee proposes that there may have been a mix-up in the calculation. Furthermore, the author has provided links for the raw human data and Neanderthal vcf, but not for the chimp sequence data. Please can this be provided? The source code is provided, but it is not sufficiently clear. For example, functions to calculate D for all conditionings should be provided and the code needs to be better documented to make clear where the other conditionings are calculated. It is quite hard to read code in a docx file, and a text file that could be loaded into a code-editor would be better. Please note that the code also contains Denisovan data which are not used in the paper. Please report in the manuscript how the code was executed to produce the results

Associate Editor: 2

Comments to the Author:
 (There are no comments.)

Comments to Author:

Reviewers' Comments to Author:

Reviewer: 1

Comments to the Author(s)

The author calculates D statistics between African and non-African modern humans, neanderthal and chimp, both on data simulated to match inferred population histories and on real data from 1001/neanderthal genome projects. When doing this, the author uses different combinations of conditioning of modern human samples on being homozygous or heterozygous at the considered variants. The author observes a strong qualitative mismatch between the results from the simulations that include neanderthal->non-African introgression and the real data and uses this to argue that this introgression event cannot explain the data. He suggests that the data is better explained by elevated mutation rates in African human populations.

First, I want to emphasise that the D-statistic as defined by Durand et al. 2011 MBE and Patterson et al. 2012 Genetics tests for significant asymmetry in four taxon topologies, as is expected under gene flow between non-sister lineages. However, the actual magnitude of the D-statistic is not easily comparable between different demographic scenarios or across tests. Hence, drawing conclusions from comparing the magnitude of different D-statistics is problematic. Connected to that, conditioning on different types of segregation patterns in the samples/populations used in the test changes the expectation of D from 0 in the absence of introgression (or other processes that could lead to $D \neq 0$) to some other value. The author is of course aware of this and bases his points on unexpected deviations from 0 when doing different conditioning. However, it is not

something the D-statistic is designed for and conclusions based on that have to be interpreted with care.

That being said, the findings by the author are unexpected, and if true interesting and worth being explored. However, what makes me skeptical about the findings is all the results shown in Fig. 1 would make perfect intuitive sense if the conditioning “ANY, HOM” etc. would mean $P1=HOM$, $P2=ANY$, rather than the other way round as it is interpreted in the text and described in the figure caption.

Such an explanation describes the data much better than the interpretation proposed by the author: That the signals are driven by increased mutation rate in Africans. In particular, only back mutations to the ancestral (chimp) state of a SNP in Africans could drive such a signal. I do think that different evolutionary rates can indeed bias Dstatistics, but, firstly, I am not sure about the intuitive explanation how this depends on conditioning, and, secondly, I do not believe that this would lead to such a strong effect among these rather recently diverged populations.

To me the most likely explanation is that the conditioning was just swapped by mistake. If this were the case then each single plot is as expected. Now they all show exactly the pattern opposite of what is expected (except for ANY/ANY and HOM/HOM were a swap would not matter). Even if we assume that there was no introgression, the at considered divergence scales segregating shared polymorphism will have much more importance than new back mutations for any realistic mutation rate. Hence, even assuming there was no introgression, the author would need to give an explanation how the expected effect of conditioning on the contribution of shared polymorphism to D-statistic, which would go into the precise opposite direction of what is observed could be totally overshadowed by back mutations.

To further elaborate on this, the expectation of D-statistics under conditioning is the following: The D-statistic is also called ABBA-BABA test, as it counts sites where you see two different alleles in $P1$ and $P2$ and two different alleles in (in this case) neanderthal and chimp. Per definition the chimp allele is called A and the neanderthal allele B.

If one conditions on $P1$ or $P2$ being heterozygous A/B then the sample being conditioned, per construction, carries the neanderthal allele and is thus expected to be closer to neanderthal. This is true even in the absence of any gene flow, if all ABBA/BABA patterns were due to ancestrally segregating polymorphism.

Hence, even in the absence of introgression, we would always expect the sample being conditioned to be het to be closer to neanderthal, or, conversely, the sample not being conditioned to be homozygous.

The author is exploring this to some degree with simulations, but I am missing simulations without gene flow and all the same comparisons as in fig. 1. For example, also considering cases with two African samples in the simulations. This would be very easy to do. My prediction is that the author would get qualitatively the same results as shown in Fig. 1 for ANY/HET and ANY/HOM if the population being conditions is swapped, i.e., for HET/ANY and HOM/ANY, etc. Only for ANY, ANY and HOM,HOM should D-statistics should be non-significant.

There needs to be a convincing explanation of how back mutations could lead to exactly to opposite observation of what is expected in a model without gene flow. What is the expected frequency of back mutations compared to segregating polymorphism?

At this stage I think it needs to be clarified whether the results are not simply due to a swap between the conditioned populations for ANY,HOM etc., which, as far as I can see, would perfectly explain all observations.

I will have further comments on more specific points but I do think the point above needs to be verified before going any further.

Author's Response to Decision Letter for (RSOS-191900.R0)

See Appendix A.

RSOS-191900.R1 (Revision)

Review form: Reviewer 1

Is the manuscript scientifically sound in its present form?

Yes

Are the interpretations and conclusions justified by the results?

No

Is the language acceptable?

Yes

Do you have any ethical concerns with this paper?

No

Have you any concerns about statistical analyses in this paper?

No

Recommendation?

Accept with minor revision (please list in comments)

Comments to the Author(s)

In manuscript the author shows that signals of elevated D-statistics in $D(\text{modern1}, \text{modern2}, \text{Neanderthal}, \text{Chimp})$ are mainly driven by heterozygous sites in Africa. Furthermore, they suggest that this effect is caused by increased mutation rate in Africans compared to non-Africans since the out-of-Africa event. The presentation of the results has considerably improved in since the last version and additional analyses strengthen the paper.

I think that the results of the analysis are highly interesting and absolutely worth publishing. The author presents a creative new statistics that is worth being explored more. The analysis seems solid now and I do not have any major critique there (but see below for one suggestion).

However, I still have one major criticism: I disagree with the confidence with which "increased mutation rate in Africa" is given as the explanation of the observed patterns. I therefore strongly suggest to tone down the confidence with which this interpretation is given as explanation. Most importantly, I would suggest to change the title as to not mention this explanation as a result. E.g., from "Signals interpreted as archaic introgression are driven primarily by accelerated evolution in Africa" to something like: "Signals interpreted as archaic introgression into out-of-Africa populations are primarily driven by rare alleles in Africans" or something along those lines, or, if "accelerated evolution" is mentioned, then to formulate it less strongly, e.g., "Signals interpreted as archaic introgression in out-of-Africa populations are (more) consistent with accelerated evolution in Africa".

The results are highly interesting as they are, even if it turns out that a different phenomenon describes the patterns, so I do not see a need to give such a strong interpretation in the title.

Increase mutation rate in African is indeed a possible explanation, that seems consistent with the data, and it is totally fine to discuss this as a possible explanation in the discussion. But as it stands this is not a result of the paper, it is an interpretation of the results.

There are alternatives to the "increased mutation rate in Africans" explanation for the patterns, which I think should be discussed in the discussion. Just from the top of my head I can think of two more but it is likely that there are more.

1) I believe that introgression from an archaic hominid lineage into Africans after the African/non-African split, that is, any lineage that is an outgroup to (modern human, Neanderthal), would be able to explain the data at least as well as the "increased mutation rate in Africans" hypothesis. This would lead to an enrichment of low-frequency (and therefore likely to be het) sites, at which Africans are more likely to carry the ancestral allele compared to non-Africans and Neanderthal. It would be very easy to simulate such a scenario with ms.

A potential complication with this explanation would be if the 1000genomes African samples were not monophyletic with respect to non-Africans (I am not sure about this, not a human geneticist). Then it would not be clear why introgression would have apparently happened into all African populations, but subsequent gene flow between African populations could be an explanation. (Also, this would similarly raise the question for the "increased mutation rate in Africans" hypothesis why all African populations have similar increased mutation rates that no non-African population has. I am not convinced environmental factors exclusive to all of Africa could explain this.)

2) A consistent difference in patterns of sequencing errors between African and non-African populations could potentially also explain (some of) the observed patterns. The data used in this analysis are imputed variant calls from low coverage sequencing. It is true that they are heavily curated, but as far as I am aware imputation does work less well in (the more diverse) Africans than in less diverse non-Africans where it is easier to impute haplotypes from reference panels. Such higher errors will be most pronounced in heterozygotes and would make Africans more different from non-Africans and Neanderthal. (Of course it would not explain non-conditioned results by others that used high-coverage data). I do not think that this is the most likely explanation, but still I suggest that the author could easily exclude it by rerunning his analysis on a few high coverage samples. There is high coverage data publicly available. I am not asking to redo everything with high coverage data, but just confirm the main results with a few well-curated high coverage samples (e.g., I think there is trio data available) and by making sure that result do not correlated with sequencing coverage.

Other general remarks:

I would suggest to properly define the D statistic on first mention, since it is so important for the results of this paper. Also, D-statistic is usually defined as proportional to ABBA-BABA (e.g., in Green et al. 2010, to which the author refers for the statistic, equation S15.1). The definition the author uses has the sign reversed, i.e., is proportional to BABA-ABBA. This does not invalidate any of the results, but it makes there interpretation unnecessarily difficult for the reader. Hence, I suggest to change to the more standard definition (preferable) or at least clarify in the text.

Figure 1: I suggest to make this easier to interpret for the reader by adding labels P1 and P2 to the x and y axes of the plots, respectively, and possibly, repeat the population labels EUR, EAS, etc. for each subplot.

Specific comments:

line 50: "measures of legacy size such as D". As previously mentioned and also acknowledged in the author's rebuttal, D is not a measure of legacy size, but a test for significant asymmetry in four taxon topologies. It is indeed often interpreted as a test for introgression, but (should) not be

interpreted as a measure of legacy size. There are other statistics, like the f_4 admixture ratio which have been suggested for that. I suggest to reformulate this.

The author says in line 72 that simulation parameters were adjusted such that $D \sim 5\%$, but then in line 84, D for unconditioned simulations is given as 3.6%. How do these two go together?

line 122: "To reduce the possibility of coding errors, I also coded the calculation of D based on a probabilistic approach". It is not clear why a probabilistic approach reduces the probability of coding errors. I would just remove the first part of this phrase, or say something like, "As an additional check ..."

I do not find the analyses in section c) (lines 155 - 173) extremely convincing or conclusive. It is fine to explore tri allelic SNPs, but this does not warrant strong concluding about (differences in) the importance of back mutations, and, as mentioned above, the interpretations from it should be more speculative. (I also note that in addition to tri-allelic sites, sites with two alleles in humans and a third allele in chimpanzees could be informative about overall double mutation rates, but this is not a suggestion for the current manuscript and I have not thought it through.)

line 156: There are previous studies that acknowledge that not only introgression but also ancestral population structure can lead to non-zero D (this can of course not explain the results of this paper). This was first suggested by Eriksson and Manica, PNAS 2012 with a simulation study, and also studies on real data (especially in non-human literature) acknowledge this. I would reformulate this sentence. E.g.: Previous studies generally assumed that the most likely source ...

line 184: "conditioning produces large D " say what you are conditioning on (one individual being heterozygous)

line 202: I do not find that the argument in line 202 why $D_{ANY,HET}(African, non-African,N,C)$ is near-zero is convincing. This argument relies on the enrichment for back mutations in non-Africans by conditioning on them being het precisely canceling out the larger number of back mutations in Africans due to suggested higher mutation rates. This is not impossible, but it is not at all obvious that these signals precisely cancel. I would reformulate this.

line 214: hat  that

lines 257-261: I am not convinced at all that the long recent neanderthal haplotypes found in remains of an early modern human (Fu et al. 2015, ref 10) could be explained by variation in mutation rate. I would suggest to remove or reformulated these lines. In the current form they might discredit an otherwise highly interesting paper.

Decision letter (RSOS-191900.R1)

07-May-2020

Dear Dr Amos:

On behalf of the Editors, I am pleased to inform you that your Manuscript RSOS-191900.R1 entitled "Signals interpreted as archaic introgression are driven primarily by accelerated evolution in Africa." has been accepted for publication in Royal Society Open Science subject to minor revision in accordance with the referee suggestions. Please find the referees' comments at the end of this email.

The reviewer is reasonably happy with your revisions and believes that we should move to publication, Notwithstanding that the reviewer raises a number of points which we would like you to consider and address before publication. Please respond in full to the remaining reviewer's comments and revise your manuscript accordingly.

- Ethics statement

- Data accessibility

<http://datadryad.org/submit?journalID=RSOS&manu=RSOS-191900.R1>

- Competing interests

- Authors' contributions

- Acknowledgements

- Funding statement

Please note that we cannot publish your manuscript without these end statements included. We have included a screenshot example of the end statements for reference. If you feel that a given

heading is not relevant to your paper, please nevertheless include the heading and explicitly state that it is not relevant to your work.

Because the schedule for publication is very tight, it is a condition of publication that you submit the revised version of your manuscript before 16-May-2020. Please note that the revision deadline will expire at 00.00am on this date. If you do not think you will be able to meet this date please let me know immediately.

on behalf of Prof Steve Brown (Subject Editor)
openscience@royalsociety.org

Associate Editor Comments to Author:

Please ensure that you fully respond to the referee's remaining comments - both in the manuscript itself and in an accompanying response to reviewer document.

Reviewer comments to Author:

Reviewer: 1

Comments to the Author(s)

In manuscript the author shows that signals of elevated D-statistics in $D(\text{modern1}, \text{modern2}, \text{Neanderthal}, \text{Chimp})$ are mainly driven by heterozygous sites in Africa. Furthermore, they suggest that this effect is caused by increased mutation rate in Africans compared to non-Africans since the out-of-Africa event. The presentation of the results has considerably improved in since the last version and additional analyses strengthen the paper.

I think that the results of the analysis are highly interesting and absolutely worth publishing. The author presents a creative new statistics that is worth being explored more. The analysis seems solid now and I do not have any major critique there (but see below for one suggestion).

However, I still have one major criticism: I disagree with the confidence with which "increased mutation rate in Africa" is given as the explanation of the observed patterns. I therefore strongly suggest to tone down the confidence with which this interpretation is given as explanation. Most importantly, I would suggest to change the title as to not mention this explanation as a result. E.g., from "Signals interpreted as archaic introgression are driven primarily by accelerated evolution in Africa" to something like: "Signals interpreted as archaic introgression into out-of-Africa populations are primarily driven by rare alleles in Africans" or something along those lines, or, if "accelerated evolution" is mentioned, then to formulate it less strongly, e.g., "Signals interpreted as archaic introgression in out-of-Africa populations are (more) consistent with accelerated evolution in Africa".

The results are highly interesting as they are, even if it turns out that a different phenomenon describes the patterns, so I do not see a need to give such a strong interpretation in the title. Increase mutation rate in African is indeed a possible explanation, that seems consistent with the data, and it is totally fine to discuss this as a possible explanation in the discussion. But as it stands this is not a result of the paper, it is an interpretation of the results.

There are alternatives to the "increased mutation rate in Africans" explanation for the patterns, which I think should be discussed in the discussion. Just from the top of my head I can think of two more but it is likely that there are more.

1) I believe that introgression from an archaic hominid lineage into Africans after the African/non-African split, that is, any lineage that is an outgroup to (modern human, Neanderthal), would be able to explain the data at least as well as the "increased mutation rate in Africans" hypothesis. This would lead to an enrichment of low-frequency (and therefore likely to be het) sites, at which Africans are more likely to carry the ancestral allele compared to non-Africans and Neanderthal. It would be very easy to simulate such a scenario with ms.

A potential complication with this explanation would be if the 1000genomes African samples were not monophyletic with respect to non-Africans (I am not sure about this, not a human geneticist). Then it would not be clear why introgression would have apparently happened into all African populations, but subsequent gene flow between African populations could be an explanation. (Also, this would similarly raise the question for the "increased mutation rate in Africans" hypothesis why all African populations have similar increased mutation rates that no non-African population has. I am not convinced environmental factors exclusive to all of Africa could explain this.)

2) A consistent difference in patterns of sequencing errors between African and non-African populations could potentially also explain (some of) the observed patterns. The data used in this analysis are imputed variant calls from low coverage sequencing. It is true that they are heavily curated, but as far as I am aware imputation does work less well in (the more diverse) Africans than in less diverse non-Africans where it is easier to impute haplotypes from reference panels. Such higher errors will be most pronounced in heterozygotes and would make Africans more different from non-Africans and Neanderthal. (Of course it would not explain non-conditioned results by others that used high-coverage data). I do not think that this is the most likely explanation, but still I suggest that the author could easily exclude it by rerunning his analysis on a few high coverage samples. There is high coverage data publicly available. I am not asking to redo everything with high coverage data, but just confirm the main results with a few well-curated high coverage samples (e.g., I think there is trio data available) and by making sure that result do not correlated with sequencing coverage.

Other general remarks:

I would suggest to properly define the D statistic on first mention, since it is so important for the results of this paper. Also, D-statistic is usually defined as proportional to ABBA-BABA (e.g., in Green et al. 2010, to which the author refers for the statistic, equation S15.1). The definition the author uses has the sign reversed, i.e., is proportional to BABA-ABBA. This does not invalidate any of the results, but it makes there interpretation unnecessarily difficult for the reader. Hence, I suggest to change to the more standard definition (preferable) or at least clarify in the text.

Figure 1: I suggest to make this easier to interpret for the reader by adding labels P1 and P2 to the x and y axes of the plots, respectively, and possibly, repeat the population labels EUR, EAS, etc. for each subplot.

Specific comments:

line 50: "measures of legacy size such as D". As previously mentioned and also acknowledged in the author's rebuttal, D is not a measure of legacy size, but a test for significant asymmetry in four taxon topologies. It is indeed often interpreted as a test for introgression, but (should) not be interpreted as a measure of legacy size. There are other statistics, like the f4 admixture ratio which have been suggested for that. I suggest to reformulate this.

The author says in line 72 that simulation parameters were adjusted such that $D \sim 5\%$, but then in line 84, D for unconditioned simulations is given as 3.6%. How do these two go together?

line 122: "To reduce the possibility of coding errors, I also coded the calculation of D based on a probabilistic approach". It is not clear why a probabilistic approach reduces the probability of coding errors. I would just remove the first part of this phrase, or say something like, "As an additional check ..."

I do not find the analyses in section c) (lines 155 - 173) extremely convincing or conclusive. It is fine to explore tri allelic SNPs, but this does not varant strong concluding about (differences in) the importance of back mutations, and, as mentioned above, the interpretations from it should be more speculative. (I also note that in addition to tri-allelic sites, sites with two alleles in humans and a third allele in chimpanzees could be informative about overall double mutation rates, but this is not a suggestion for the current manuscript and I have not thought it through.)

line 156: There are previous studies that acknowledge that not only introgression but also ancestral population structure can lead to non-zero D (this can of course not explain the results of this paper). This was first suggested by Eriksson and Manica, PNAS 2012 with a simulation study, and also studies on real data (especially in non-human literature) acknowledge this. I would reformulate this sentence. E.g.: Previous studies generally assumed that the most likely source ...

line 184: "conditioning produces large D" say what you are conditioning on (one individual being heterozygous)

line 202: I do not find that the argument in line 202 why DANY,HET(African, non-African,N,C) is near-zero is convincing. This argument relies on the enrichment for back mutations in non-Africans by conditioning on them being het precisely canceling out the larger number of back mutations in Africans due to suggested higher mutation rates. This is not impossible, but it is not at all obvious that these signals precisely cancel. I would reformulate this.

line 214: hat  that

lines 257-261: I am not convinced at all that the long recent neanderthal haplotypes found in remains of an early modern human (Fu et al. 2015, ref 10) could be explained by variation in mutation rate. I would suggest to remove or reformulated these lines. In the current form they might discredit an otherwise highly interesting paper.

Author's Response to Decision Letter for (RSOS-191900.R1)

See Appendix B.

Decision letter (RSOS-191900.R2)

12-Jun-2020

Dear Dr Amos,

It is a pleasure to accept your manuscript entitled "Signals interpreted as archaic introgression appear to be driven primarily by accelerated evolution in Africa." in its current form for publication in Royal Society Open Science. The comments of the reviewer(s) who reviewed your manuscript are included at the foot of this letter.

on behalf of Steve Brown (Subject Editor)
openscience@royalsociety.org

Associate Editor Comments to Author:

Comments to the Author:

The author has engaged with the concerns of the reviewer, and also provided an enhanced conflict of interest declaration, providing some reassurance regarding this. Given the reviewer recommended acceptance once the scientific changes had been made, the paper may be accepted as is.

Appendix A

General response. The primary criticism is that the Referee believes that I may have coded the conditioning the wrong way round. I have checked this and do not believe I have. Moreover, several of the most striking results are robust to getting the conditioning inverted. Tellingly, in comparisons between pairs of Africans, reversing the individuals has little impact on conditioned D, a new analysis that is included in the revised text. In addition, I have added extra simulations as requested by the Referee. Below is a detailed, point-by-point rebuttal of all points raised. I think the only point I have not addressed is the request to provide proof that mutation rate differences exist and can explain the observed values of D. There are, in fact, several papers that present evidence for variation in mutation rate that the Referee overlooks / ignores. This would be a major and challenging piece of work that I believe goes well beyond the intended scope of the current paper. However, I have added an analysis showing that back-mutations are far commoner than is usually assumed.

Comments to the Author(s)

The author calculates D statistics between African and non-African modern humans, neanderthal and chimp, both on data simulated to match inferred population histories and on real data from 1001/neanderthal genome projects. When doing this, the author uses different combinations of conditioning of modern human samples on being homozygous or heterozygous at the considered variants. The author observes a strong qualitative mismatch between the results from the simulations that include neanderthal->non-African introgression and the real data and uses this to argue that this introgression event cannot explain the data. He suggests that the data is better explained by elevated mutation rates in African human populations.

First, I want to emphasise that the D-statistic as defined by Durand et al. 2011 MBE and Patterson et al. 2012 Genetics tests for significant asymmetry in four taxon topologies, as is expected under gene flow between non-sister lineages. However, the actual magnitude of the D-statistic is not easily comparable between different demographic scenarios or across tests.

- **Response:** I could not agree more! However, here I am using D as a broadly qualitative measure, asking whether values are either large or near-zero. I deliberately and, I believe, correctly, never try to interpret the values in terms of the inferred level of introgression.

Hence, drawing conclusions from comparing the magnitude of different D-statistics is problematic. Connected to that, conditioning on different types of segregation patterns in the samples/populations used in the test changes the expectation of D from 0 in the absence of introgression (or other processes that could lead to $D \neq 0$) to some other value. The author is of course aware of this and bases his points on unexpected deviations from 0 when doing different conditioning. However, it is not something the D-statistic is designed for and conclusions based on that have to be interpreted with care.

- **Response:** Broadly, I agree: D is a rather pragmatic quantification of asymmetrical base-sharing that has a number of issues, not least its absolute assumption that mutation rate is constant and the fact that quantitative interpretation requires accurate knowledge of when taxa split. However, its simplicity is also its strength, in that it is usually easy to predict when D should be positive, zero or negative under different scenarios.

While I agree that care is needed, I would point out that my study has excellent internal controls. First, I show that conditioning alone does not create an artefactual signal because $D \sim 0$ for all conditioned comparisons between non-Africans, a pattern confirmed by simulation. Thus, Africans must be in some way different. I then show that conditioning pairs of Africans always produces large D and that the sign of D does not depend on which way round the two individuals are. Logically, this shows that all Africans carry

the signal a signal capable of generating large D that is exposed by conditioning. Finally, African – non-African comparisons are fully consistent with this model: conditioning the non-African has little effect but conditioning Africans to be homozygous kills the signal.

That being said, the findings by the author are unexpected, and if true interesting and worth being explored. However, what makes me skeptical about the findings is all the results shown in Fig. 1 would make perfect intuitive sense if the conditioning “ANY, HOM” etc. would mean $P1=HOM$, $P2=ANY$, rather than the other way round as it is interpreted in the text and described in the figure caption.

- **Response:** I interpret ‘make perfect intuitive sense’ to mean ‘consistent with introgression’. However, this idea works only for African – non-African comparisons. In all other cases introgression does not explain the pattern I report. Thus, under an introgression model:

1. Conditioning should only generate large D between non-Africans, but the observation is that D is always near-zero.
2. Condition should have little or no effect inside Africa, but the observation is that D is always large in conditioned African-African pairs.
3. Conditioning, if dependent on the presence of introgressed fragments, should reverse its sign when the two individuals being compared are reversed, but the observation is that D is little changed.

More pragmatically, I have supplied my code which the Referee is welcome to inspect. I have re-checked this code and cannot see any obvious error though we are all human so I hope the code is clear enough for the Referee also to check that I have not made an error. I would of course be delighted if the Referee replicated my analysis as an extra check using their own code.

Note, I acknowledge that recent papers are emerging that report appreciable legacies in Africans. Nonetheless, if we put trust in the early introgression papers it should still be true that non-Africans carry relatively much more. Consequently, my arguments still hold, even if the values for conditioned D within African may, under these circumstances, be non-zero – conditioning should always produce a larger effect outside Africa where the classic studies indicate legacies to be greatest.

Such an explanation describes the data much better than the interpretation proposed by the author: That the signals are driven by increased mutation rate in Africans.

- **Response:** I could not disagree more strongly. As I explain above, I believe my analysis is robust to simple programming errors and clearly demonstrates that the signal picked up by D is driven almost entirely by heterozygous sites in Africans causing unexpectedly high divergence from archaics. The analysis is transparent and easy to replicate by anyone who has the inclination to do so.

In particular, only back mutations to the ancestral (chimp) state of a SNP in Africans could drive such a signal.

- **Response:** indeed. In the 1000 genomes data there are ~250,000 sites carrying three alleles, implying two mutations at the same site within humans, one of which must be a transversion. Since transversions are at least two times less likely than transitions, the number of sites carrying back-mutations to the ancestral state will be approximately 500,000. Since back-mutations will tend to be recent they will also tend to be rare and therefore will appear mostly in the heterozygous state. I have encountered comments that tri-allelic sites are ‘likely most due to sequencing errors’ but this would suggest that the 1000 genome data have been atrociously curated and near-worthless. Moreover, only about 1% of tri-allelic sites carry

an allele present as only a single copy, a very large majority cannot reasonably be dismissed as sequencing errors. This analysis is now added to the revised text.

I do think that different evolutionary rates can indeed bias D statistics, but, firstly, I am not sure about the intuitive explanation how this depends on conditioning, and, secondly, I do not believe that this would lead to such a strong effect among these rather recently diverged populations.

- Response: I don't know what to say beyond inviting the Referee to try to replicate what I have done. If I have made a mistake I will gladly accept it and I have no wish at all to publish something that will later be ridiculed because of some simple mistake. However, I have recoded this analysis twice and, for the reasons given above, I have high confidence that what I report and my interpretation are both correct.

To me the most likely explanation is that the conditioning was just swapped by mistake. If this were the case then each single plot is as expected.

- Response: see above.

Now they all show exactly the pattern opposite of what is expected (except for ANY/ANY and HOM/HOM where a swap would not matter). Even if we assume that there was no introgression, the at considered divergence scales segregating shared polymorphism will have much more importance than new back mutations for any realistic mutation rate.

- Response: this assertion is incorrect because D is a normalised difference. The more recent the split, the more segregating sites due to incomplete lineage sorting will be present and hence the larger any absolute asymmetry needs to be to achieve a given D. Conversely, the longer ago humans and Neanderthals split, the fewer the number of segregating sites and even modest asymmetric ABBAs and BABAs will be able to generate large D. Green et al. chose a divergence time of 300,000 years ago which is far more recent than most anthropologists believe. I went to a recent seminar by Chris Stringer who argued that the split must have been at least 500,000 years ago and likely towards 700,000, a value where both simulation and the theory developed by Durand et al. show that segregating sites will be much rarer and hence that small asymmetries would generate much larger values of D.

Hence, even assuming there was no introgression, the author would need to give an explanation how the expected effect of conditioning on the contribution of shared polymorphism to D-statistic, which would go into the precise opposite direction of what is observed could be totally overshadowed by back mutations.

- Response: I am sorry but I fail to understand this statement fully. I think the point being raised is that shared polymorphisms, when conditioned, will generate non-zero D. This is shown not to be the case in both directly and in theory. If the suggestion were to be correct, since all humans carry shared polymorphisms, all comparisons should yield non-zero D. In fact, all comparisons outside Africa yield $D \sim 0$ however conditioning is applied. Second, shared polymorphisms are old enough such that the mean frequency of derived alleles will be near 50%. Consequently, there is no inherent asymmetry for the conditioning process to expose / emphasise.

To further elaborate on this, the expectation of D-statistics under conditioning is the following: The D-statistic is also called ABBA-BABA test, as it counts sites where you see two different alleles in P1 and P2 and two different alleles in (in this case) neanderthal and chimp. Per definition the chimp allele is called A and the neanderthal allele B.

If one conditions on P1 or P2 being heterozygous A/B then the sample being conditioned, per construction, carries the neanderthal allele and is thus expected to be closer to neanderthal. This is true even in the absence of any gene flow, if all ABBA/BABA patterns were due to ancestrally segregating polymorphism.

Hence, even in the absence of introgression, we would always expect the sample being conditioned to be het to be closer to neanderthal, or, conversely, the sample not being conditioned to be homozygous.

- **Response:** See previous point. Conditioning on one sample being heterozygote does indeed guarantee that a Neanderthal allele is carried. However, non-zero D will only result if the second individual being compared is either more or less likely to be homozygous BB. If the two alleles are equally likely to rare / common, on average the second individual is as likely to BB as it is to be AA. Conditioning can only have an effect on alleles that are recent enough not to have approximated the equilibrium state of having a meta frequency $\sim 50\%$.

The author is exploring this to some degree with simulations, but I am missing simulations without gene flow and all the same comparisons as in fig. 1. For example, also considering cases with two African samples in the simulations. This would be very easy to do. My prediction is that the author would get qualitatively the same results as shown in Fig. 1 for ANY/HET and ANY/HOM if the population being conditions is swapped, i.e., for HET/ANY and HOM/ANY, etc. Only for ANY, ANY and HOM,HOM should D-statistics should be non-significant.

- **Response:** I have added these extra simulations as requested. In the absence of introgression, D never departs significantly from zero.

There needs to be a convincing explanation of how back mutations could lead to exactly to opposite observation of what is expected in a model without gene flow. What is the expected frequency of back mutations compared to segregating polymorphism?

- **Response:** I believe this request goes massively beyond the scope of the current paper. I use previous published work in high profile journals as my reference. Let us assume there are three possible models capable of generating non-zero D:

a) Introgression

b) Back-mutations in a scenario where populations differ in mutation rate

c) Some other, as yet unknown explanation

Previous work all focus only on (a), invariably assuming without evidence or justification that (b) is false and never considering (c). In contrast, my analysis seeks to separate (a) and (b) using the conditioning method. As I argue, I believe the results are logically incompatible with (a) and therefore must prove (b) unless there is a third option, so far not considered by anyone. Far from requiring additional evidence for mutation rate differences (quite a lot already exists, though is largely ignored by those reporting introgression), I feel that the onus is now strongly on those favouring introgression either to show that I have made a big mistake in my analyses or to identify a flaw in my logic.

At this stage I think it needs to be clarified whether the results are not simply due to a swap between the conditioned populations for ANY,HOM etc., which, as far as I can see, would perfectly explain all observations.

I will have further comments on more specific points but I do think the point above needs to be verified before going any further.

Appendix B

Comments to the Author(s)

In manuscript the author shows that signals of elevated D-statistics in $D(\text{modern1}, \text{modern2}, \text{Neanderthal}, \text{Chimp})$ are mainly driven by heterozygous sites in Africa. Furthermore, they suggest that this effect is caused by increased mutation rate in Africans compared to non-Africans since the out-of-Africa event. The presentation of the results has considerably improved in since the last version and additional analyses strengthen the paper.

I think that the results of the analysis are highly interesting and absolutely worth publishing. The author presents a creative new statistics that is worth being explored more. The analysis seems solid now and I do not have any major critique there (but see below for one suggestion).

However, I still have one major criticism: I disagree with the confidence with which "increased mutation rate in Africa" is given as the explanation of the observed patterns. I therefore strongly suggest to tone down the confidence with which this interpretation is given as explanation. Most importantly, I would suggest to change the title as to not mention this explanation as a result. E.g., from "Signals interpreted as archaic introgression are driven primarily by accelerated evolution in Africa" to something like: "Signals interpreted as archaic introgression into out-of-Africa populations are primarily driven by rare alleles in Africans" or something along those lines, or, if "accelerated evolution" is mentioned, then to formulate it less strongly, e.g., "Signals interpreted as archaic introgression in out-of-Africa populations are (more) consistent with accelerated evolution in Africa".

Response: I rather disagree with the Referee, in that the other models suggested don't really work. Nonetheless, I have diluted the meaning by adding 'appear to be'.

The results are highly interesting as they are, even if it turns out that a different phenomenon describes the patterns, so I do not see a need to give such a strong interpretation in the title. Increase mutation rate in African is indeed a possible explanation, that seems consistent with the data, and it is totally fine to discuss this as a possible explanation in the discussion. But as it stands this is not a result of the paper, it is an interpretation of the results.

There are alternatives to the "increased mutation rate in Africans" explanation for the patterns, which I think should be discussed in the discussion. Just from the top of my head I can think of two more but it is likely that there are more.

1) I believe that introgression from an archaic hominid lineage into Africans after the African/non-African split, that is, any lineage that is an outgroup to (modern human, Neanderthal), would be able to explain the data at least as well as the "increased mutation rate in Africans" hypothesis. This would lead to an enrichment of low-frequency (and therefore likely to be het) sites, at which Africans are more likely to carry the ancestral allele compared to non-Africans and Neanderthal. It would be very easy to simulate such a scenario with ms.

A potential complication with this explanation would be if the 1000genomes African samples were not monophyletic with respect to non-Africans (I am not sure about this, not a human geneticist). Then it would not be clear why introgression would have apparently happened into all African populations, but subsequent gene flow between African populations could be an explanation. (Also, this would similarly raise the question for the "increased mutation rate in Africans" hypothesis why all African populations have similar increased mutation rates that no non-African population has. I am not convinced environmental factors exclusive to all of Africa could explain this.)

2) A consistent difference in patterns of sequencing errors between African and non-African populations could potentially also explain (some of) the observed patterns. The

data used in this analysis are imputed variant calls from low coverage sequencing. It is true that they are heavily curated, but as far as I am aware imputation does work less well in (the more diverse) Africans than in less diverse non-Africans where it is easier to impute haplotypes from reference panels. Such higher errors will be most pronounced in heterozygotes and would make Africans more different from non-Africans and Neanderthal. (Of course it would not explain non-conditioned results by others that used high-coverage data). I do not think that this is the most likely explanation, but still I suggest that the author could easily exclude it by rerunning his analysis on a few high coverage samples. There is high coverage data publicly available. I am not asking to redo everything with high coverage data, but just confirm the main results with a few well-curated high coverage samples (e.g., I think there is trio data available) and by making sure that result do not correlated with sequencing coverage.

Response: I have added an extra paragraph in the discussion where I mention alternative models, including population sub-structure. I do not mention sequencing errors any more because it really would be

Other general remarks:

I would suggest to properly define the D statistic on first mention, since it is so important for the results of this paper. Also, D-statistic is usually defined as proportional to ABBA-BABA (e.g., in Green et al. 2010, to which the author refers for the statistic, equation S15.1). The definition the author uses has the sign reversed, i.e., is proportional to BABA-ABBA. This does not invalidate any of the results, but it makes there interpretation unnecessarily difficult for the reader. Hence, I suggest to change to the more standard definition (preferable) or at least clarify in the text.

Response: text adjusted in line with the suggestion.

Figure 1: I suggest to make this easier to interpret for the reader by adding labels P1 and P2 to the x and y axes of the plots, respectively, and possibly, repeat the population labels EUR, EAS, etc. for each subplot.

Response: changes made as suggested.

Specific comments:

line 50: "measures of legacy size such as D". As previously mentioned and also acknowledged in the author's rebuttal, D is not a measure of legacy size, but a test for significant asymmetry in four taxon topologies. It is indeed often interpreted as a test for introgression, but (should) not be interpreted as a measure of legacy size. There are other statistics, like the f4 admixture ratio which have been suggested for that. I suggest to reformulate this.

Response: indeed! Text clarified as requested.

The author says in line 72 that simulation parameters were adjusted such that $D \sim 5\%$, but then in line 84, D for unconditioned simulations is given as 3.6%. How do these two go together?

Response: this was a typo. I have run a lot of these simulations and used to use slightly different parameter values. I have corrected the expected value to $\sim 4\%$.

line 122: "To reduce the possibility of coding errors, I also coded the calculation of D based on a probabilistic approach". It is not clear why a probabilistic approach reduces the probability of coding errors. I would just remove the first part of this phrase, or say something like, "As an additional check ..."

Response: suggested correction added.

I do not find the analyses in section c) (lines 155 - 173) extremely convincing or conclusive. It is fine to explore tri allelic SNPs, but this does not varant strong concluding about (differences in) the importance of back mutations, and, as mentioned above, the

interpretations from it should be more speculative. (I also note that in addition to tri-allelic sites, sites with two alleles in humans and a third allele in chimpanzees could be informative about overall double mutation rates, but this is not a suggestion for the current manuscript and I have not thought it through.)

Response: I tend to agree. However, there are many out there who assert, without proper thought, that the back-mutation rate is far too low to be a concern. This assertion appears based on the naïve assumption that mutation rate is constant (a crude back-of-envelope calculation shows an expectation of under 100 triallelic sites across the genome!). I do not pretend this calculation is accurate but I do think it is needed to counter the correct assumption that if back-mutations are rare, my model does not work. I have added text to clarify.

line 156: There are previous studies that acknowledge that not only introgression but also ancestral population structure can lead to non-zero D (this can of course not explain the results of this paper). This was first suggested by Eriksson and Manica, PNAS 2012 with a simulation study, and also studies on real data (especially in non-human literature) acknowledge this. I would reformulate this sentence. E.g.: Previous studies generally assumed that the most likely source ...

Response: suggested correction added.

line 184: "conditioning produces large D" say what you are conditioning on (one individual being heterozygous)

Response: wording clarified.

line 202: I do not find that the argument in line 202 why D_{ANY,HET}(African, non-African,N,C) is near-zero is convincing. This argument relies on the enrichment for back mutations in non-Africans by conditioning on them being het precisely canceling out the larger number of back mutations in Africans due to suggested higher mutation rates. This is not impossible, but it is not at all obvious that these signals precisely cancel. I would reformulate this.

Response: this is a tricky concept to get across. I have reworded the text to make it clearer. Basically, the conditioning acts to exclude all sites that would otherwise carry a positive signal.

line 214: hat  that

Response: corrected

lines 257-261: I am not convinced at all that the long recent neanderthal haplotypes found in remains of an early modern human (Fu et al. 2015, ref 10) could be explained by variation in mutation rate. I would suggest to remove or reformulated these lines. In the current form they might discredit an otherwise highly interesting paper.

Response: I have added text to make it clear that I exclude genuinely hybrid skeletons from this comment.